# Primary Founder Mutations in the *PRKDC* Gene Increase Tumor Mutation Load in Colorectal Cancer

**DOI:** 10.3390/ijms23020633

**Published:** 2022-01-06

**Authors:** Hajnalka Laura Pálinkás, Lőrinc Pongor, Máté Balajti, Ádám Nagy, Kinga Nagy, Angéla Békési, Giampaolo Bianchini, Beáta G. Vértessy, Balázs Győrffy

**Affiliations:** 1Genome Metabolism Research Group, Institute of Enzymology, Research Centre for Natural Sciences, Magyar Tudósok Körútja 2, H-1117 Budapest, Hungary; palinkas.hajnalka@ttk.hu (H.L.P.); nagy.kinga@ttk.hu (K.N.); bekesi.angela@ttk.hu (A.B.); 2Department of Applied Biotechnology and Food Sciences, BME Budapest University of Technology and Economics, Szt Gellért tér 4, H-1111 Budapest, Hungary; 3TTK Lendület Cancer Biomarker Research Group, Institute of Enzymology, Research Centre for Natural Sciences, Magyar Tudósok Körútja 2, H-1117 Budapest, Hungary; pongorlorinc@gmail.com (L.P.); balajtimate@gmail.com (M.B.); nagy.adam@ttk.hu (Á.N.); 4Department of Bioinformatics and 2nd Department of Pediatrics, Semmelweis University, Tűzoltó u. 7-9, H-1094 Budapest, Hungary; 5Department of Medical Oncology, San Raffaele Scientific Institute, via Olgettina 60, 20132 Milan, Italy; bianchini.giampaolo@hsr.it

**Keywords:** cancer, DNA repair, mutation burden, non-homologous end joining, survival, next generation sequencing

## Abstract

The clonal composition of a malignant tumor strongly depends on cellular dynamics influenced by the asynchronized loss of DNA repair mechanisms. Here, our aim was to identify founder mutations leading to subsequent boosts in mutation load. The overall mutation burden in 591 colorectal cancer tumors was analyzed, including the mutation status of DNA-repair genes. The number of mutations was first determined across all patients and the proportion of genes having mutation in each percentile was ranked. Early mutations in DNA repair genes preceding a mutational expansion were designated as founder mutations. Survival analysis for gene expression was performed using microarray data with available relapse-free survival. Of the 180 genes involved in DNA repair, the top five founder mutations were in *PRKDC* (*n* = 31), *ATM* (*n* = 26), *POLE* (*n* = 18), *SRCAP* (*n* = 18), and *BRCA2* (*n* = 15). *PRKDC* expression was 6.4-fold higher in tumors compared to normal samples, and higher expression led to longer relapse-free survival in 1211 patients (HR = 0.72, *p* = 4.4 × 10^−3^). In an experimental setting, the mutational load resulting from UV radiation combined with inhibition of *PRKDC* was analyzed. Upon treatments, the mutational load exposed a significant two-fold increase. Our results suggest *PRKDC* as a new key gene driving tumor heterogeneity.

## 1. Introduction

Following DNA damage, healthy human cells activate signaling cascades to prevent cell-cycle progression, as well as to initiate repair mechanisms through DNA damage response. Once the accumulated damage is beyond repair, apoptosis is induced [1]. The main mechanisms of DNA repair include base-excision repair (BER), mismatch repair (MMR), nucleotide-excision repair (NER), homologous recombination (HR), and non-homologous end joining (NHEJ).

NER is utilized after UV radiation-induced injury, where the resulting DNA adducts are recognized, and the damaged, short, single strand is removed and resynthesized by a DNA polymerase using the unharmed strand as the template. BER is involved in repairing non-helix distorting base lesions. These lesions are recognized by DNA glycosylases, which remove damaged and inappropriate bases, forming abasic AP sites, cleaved by AP endonucleases. The single-strand break is then repaired by either short- or long-patch BER. MMR recognizes errors in the course of cell replication, where mismatch repair complexes repair errors occurring on the daughter strand using the parental strand as the template. After double-strand breaks, the damage is either repaired by homologous recombination during S and G2 phases, which enables the repair of the damage, or by NHEJ, where ends with microhomologies are joined. A comprehensive review of DNA repair mechanisms was presented previously [2].

One of the major differences between normal and cancer cells is their response to DNA damage and repair. In normal cells, blocking DNA replication at lesions during double-strand repair leads to the collapse of the replication fork. The collapsed fork is then recognized by ATM and ATR proteins, which, by signaling through TP53, induce apoptosis by provoking BAX translocation and interaction to the mitochondrial voltage-dependent anion channel. In tumor cells, apoptotic pathways are suppressed by mutations in oncogenes (most commonly TP53) [3]. Cancer cells frequently have somatic mutations in DNA-repair genes (*ATM*, *BRCA1/2*, and *FANC* genes) that decrease repair capabilities, leading to the accumulation of mutations. The effect is further enhanced in the case of TP53 double mutants, showing a relationship between DNA repair defects and TP53-induced apoptosis [4].

Our aim was to identify founder mutation events by evaluating a large panel of colon cancer samples analyzed via next generation sequencing. Here, a founder mutation is designated as a genetic variation occurring when a new population is established—in terms of cancer etiology, such a novel population is always based on a clonal expansion of the cancer cells. By comparing mutation frequency, proportion, and estimated mutation time, we aimed to identify founder events leading to increased tumor heterogeneity. In the second part of the study, by utilizing a cell culture framework, we set the goal to validate the presence of a higher mutation load once a founder mutation occurred. Finally, by evaluating a large panel of independent clinical samples, we assessed the clinical relevance of our findings.

## 2. Results

### 2.1. DNA-Repair Gene Defects Cause an Increased Accumulation Rate of Mutations

We investigated mutation distribution in 591 colorectal cancer cases from the TCGA repository (see details in Section 4.2). Mutation burden (total mutation count) was calculated for each sample (for experimental workflow, see Figure 1A). In the case of mutation in any of the DNA repair mechanisms, a significant increase (*p* < 10^−16^) was observed in the overall mutation burden compared to the wild-type samples (Figure 1B). We also investigated whether a mutation in a DNA repair pathway will result in increased mutation prevalence in another pathway as well. Multiple mutations were not observed in 49.4% of the samples (25.8% had no mutation, while 23.6% had one mutation in a DNA repair pathway). However, once a mutation in one of the pathways occurred, another hit in an alternate pathway was more common (15.3–32.4% pairwise for all cases, 7.42 × 10^−13^ > *p* > 3.58 × 10^−34^) (Figure 1C).

### 2.2. Identifying Genes Where the Mutation Status Is Associated with High Mutation Burden

Calculation of mutation burden was performed to evaluate the effect of mutation in each DNA-repair gene and across three independent datasets (namely TCGA, DFCI, and Genentech repositories, see details in Section 4.2). When looking at the top genes in all three datasets (Table 1), three genes (*ATM*, *BRCA2*, and *PRKDC*) were recurrently identified. Samples with DNA-repair mutations in the top genes had a 6–12× increase in median mutation burden compared to wild-type samples (Figure 1D).

### 2.3. Founder Mutation and Increased Mutational Load

An important aspect of mutations is not only the sole presence of a mutation, but also its ability to lead to a “mutator” phenotype. To investigate this issue, we identified DNA-repair gene mutations where the mutation developed early (this feature is proven by the high proportion of mutant reads when compared to other mutations) and the bulk of the accumulated mutations occurred later (manifested in lower mutation proportions). Figure 2A summarizes the concept of the analysis. Mutations with high prevalence preceding these lower mutation frequency peaks of mutations were defined as potential founder gene mutations. A total of 125 patients from the TCGA dataset (see details in Materials and Methods) had such a potential founder mutation—35.2% of these resulted in a single accumulation peak (Figure 2B), 21.6% led to double accumulation peaks (Figure 2C), and 14.4% delivered an exponential mutation accumulation (Figure 2D).

Following a DNA-repair gene mutation, one would expect that mutation rates would increase at similar (proximal) frequencies. Interestingly, DNA-repair mutations did not always lead to this effect, as in many cases the bulk of “randomly” accumulated mutations had a mean frequency difference over 40% compared to the DNA-repair gene defect. However, we have to note that these were patients with very few mutations.

### 2.4. PRKDC Acting as a Mutator Phenotype

We restricted the founder genes to cases where at least half of the randomly accumulated mutations were in a mutation frequency range of 30% after the DNA-repair gene defect. Since one sample might have several DNA-repair genes that passed this criterion, we selected the top three genes for each sample. In this analysis, the top three most common genes with founder mutations were *PRKDC* (*n* = 31 colorectal cancer samples from the TCGA dataset), *ATM* (*n* = 18 samples), and *BRCA2* (*n* = 17 samples). The median mutation frequency of *PRKDC* founder mutations was 69.5% in these patients (range 66.6%–98.9%). Most of the mutations were classified as missense variants, frame shift insertions, and deletions (Figure 3).

### 2.5. Mutagenesis Experiment

The experimental workflow of the mutagenesis experiments is summarized in Figure 4. For the details of sample collection, please see the Materials and Methods. Altogether, 16 samples were whole-exome sequenced following the mutagenic and inhibitory treatments. These include each treatment (iPRKDC, iATM, and the combination of the two) in triplicates, and the normal control in quadruplicate. The mean number of sequenced reads ranged between 30 and 50 million. All samples fulfilled the minimal quality requirement of at least 25 million reads, with a minimum coverage of 75×. To improve the quality of reads, a trimming was performed, which resulted in the increase of mean quality scores to 33. The alignment of sequences to the reference genome resulted in the placement of 99% of sequences.

### 2.6. Effect of PRKDC and ATM1 Inhibition on Mutation Rate in UV-Treated Cell Lines

The average number of mutations in the samples after UV exposure was 9151. This number showed an increase to 9960 in the iPRKDC-treated cell lines (*p* = 0.042) and to a mean of 9681 in the iATM cell lines (*p* = 0.85). The combined treatment of both PRKDC and ATM inhibitors resulted in almost double the number of mutations, to an average of 18,193 (*p* = 0.00049) compared to the samples treated with UV radiation only (Figure 5). 

Of note, high impact mutations of either *PRKDC*, *ATM*, or both genes were also found to be present in essentially all of the samples treated with both *PRKDC* and *ATM* inhibitors. Around half of these mutations included the acquisition of a stop codon, resulting in a truncated protein. Mutations in other genes with a potentially mutator phenotype, such as *POLE*, were also present in the samples treated with both inhibitors.

### 2.7. Expression of DNA-Repair Genes and Survival Differences

To address the correlation between survival data and the expression levels of the DNA repair factors (especially those that were selected as top candidates for founder mutators—see Table 1), the microarray gene expression data of 2110 colorectal cancer patients were collected from NCBI Gene Expression Omnibus (GEO) database and analyzed, as described in detail in Section 4.8. Higher expression of the signatures representing each DNA repair pathway in colorectal cancer patients led to longer relapse-free survival (Appendix A). In the case of *PRKDC* and *BRCA2*, higher expression resulted in significantly better survival (HR = 0.72, *p* = 0.0044; and HR = 0.73 and *p* = 0.0056 for *PRKDC* and *BRCA2*, respectively, see Appendix A). The survival analysis using the mean expression of all DNA-repair genes resulted in the strongest correlation to improved survival (Appendix A). When comparing expression levels of tumor and normal samples, expression of *PRKDC*, *BRCA2*, and *ATM* genes was higher in the tumors. Fold-change increase of expression was over six-fold for *PRKDC* (Figure 3C), four-fold for *BRCA2*, and over 1.4× for *ATM*. The survival analysis results are listed in Table 2.

## 3. Discussion

In this study, by combining in silico analysis and in vitro experiments, we identified mutations as founder events in hypermutating colorectal tumors. The recurrent genes affected by somatic mutations were *PRKDC*, *ATM*, *SCRAP*, *BRCA2*, and *POLE*. Patients with the highest mutation burden had mutations affecting multiple repair pathways. When examining mutation frequencies in patients, we identified four distinct accumulation patterns: neutral, single expansion, double/multiple expansions, and exponential expansion. In the case of non-neutral patterns, the most prevalent mutation harbored at the beginning of expansion was in the *PRKDC* gene. 

The PRKDC protein (DNA-dependent protein kinase catalytic subunit, also abbreviated as DNA-PKcs) is involved in the DNA repair of double-strand DNA breaks [5,6] and V(D)J recombination [7,8] by non-homologous end joining. As the catalytic subunit of DNA-dependent protein kinase (DNA-PK), it is essential for the NHEJ process. Subsequent to DNA damage, a heterodimer comprised of Ku70 and Ku80 bind to the free DNA end, where PRKDC is recruited. In the case of a double-strand break, initial repairs are attempted by NHEJ, followed by HR [9,10]. During the repair process, DNA-PK functions as a “gatekeeper” by regulating DNA access through an autophosphorylation event [11]. By blocking autophosphorylation, DNA becomes inaccessible to the double-strand break repair complex, resulting in impaired HR [12,13]. 

Inhibition of PRKDC leads to enhanced cytotoxicity of radiotherapy treatment [14] and alkylating agents in cancer patients [15]. Mutations of PRKDC were significantly associated with a higher mutation load and response to immunotherapy in multiple cancer types [16]. In most responder cases, either truncating mutations or mutations in functional domains were identified. In addition, knockout of PRKDC enhanced the efficacy of the anti-programmed cell death protein one in the CT26 animal model, suggesting it as a drug target for immune checkpoint inhibitors [17]. Knockdown of PRKDC using small interfering RNA increased the sensitivity of malignant melanoma cells to cisplatin treatment [18]. Inhibition of PRKDC in osteosarcoma cell lines increased radiosensitivity, while co-treatment with the PRKDC inhibitor KU60648 resulted in enhanced DNA damage [19]. *PRKDC* mutant mice were unable to repair double-strand DNA breaks induced by ionizing radiation [20], resulting in shorter survival times.

On the other hand, the effects of elevated PRKDC expression are contradictory. In advanced prostate tumors, PRKDC is highly activated, promoting progression and metastasis [21]. In the case of breast cancer, higher expression levels of PRKDC were significantly associated with shorter overall survival, higher tumor grade, and positive lymph node status in patients, while downregulation sensitized MCF-7 cell lines to chemotherapeutics in vitro and in xenograft models [22]. In contrast, low protein expression of PRKDC was associated with a higher tumor grade and mitotic index, as well as survival, in breast cancer [23]. Here, we found that high expression of PRKDC resulted in better survival in colorectal cancer. The elevated PRKDC expression in tumors compared to normal samples may be a response to either DNA damage or increased replication. Tumors with higher PRKDC expression may have more efficient DNA damage repair compared to low expressing tumors, while mutations in PRKDC will probably decrease or inhibit its DNA repair activity.

Recent studies have shown that inhibition or deletion and loss of function of PRKDC is compensated by hyperactivation of ATM [24,25], showing an interplay between pathways [26]. The three-dimensional structure of the PRKDC protein suggests competition of binding between Ku80 and BRCA1, and activation of NHEJ and homologous recombination [27], resulting in enhanced targeting of patients with BRCA1 and BER deficiency [28]. By performing mutational load analysis in HCT116 cells, we found that combined inhibition of both PRKDC and ATM proteins, together with UV radiation, led to a significantly increased number of mutations as compared to the radiation effect on its own.

We have to mention some limitations of our study: first, the number of colorectal cancer patients with next generation sequencing data was a restrictive factor. While PRKDC was the top hit with the highest frequency, the actual number of patients was still low. Thus, a future independent study must be performed with a larger patient number to validate our findings. Second, for the in cell validation of the impact of PRKDC and ATM, we used HCT116 cells that were deficient in mismatch repair that might affect the results. However, mismatch repair deficiency frequently occurs among colorectal cancers [29,30,31] and, in this sense, the HCT116 cell line is a well-established and relevant colorectal cancer model. Still, comparative studies on the MMR-proficient version of HCT116 cells or other cell lines is among our future goals. Third, the overall effect of DNA-repair gene defects was, to a certain degree, limited—this suggests that other factors could also have a significant influence on the mutation load, in addition to DNA repair. Here, we considered the possibility of increased DNA editing activity that might significantly contribute to the increased mutational burden during cancer progression [32,33,34,35], even if such events are often episodic, and cells tend to suppress such activities in long term [36]. However, even under such increased DNA editing conditions, the DNA repair capacity of the cells must have an impact. With intact DNA repair, the edited bases often can efficiently be repaired, and not cause increased mutational burden. In this sense, even if a DNA-repair gene mutation appears as a result of a temporary increased DNA editing activity (or due to other DNA damaging effect like oxidative stress or chemotherapy), such mutation can significantly contribute to the following expansion of the mutational load, and therefore might be considered as a founder mutation.

In summary, the most common founder mutation in DNA-repair genes leading to higher subsequent mutation load in colorectal cancer was in PRKDC, a gene involved in non-homologous end joining repair. Our results suggest PRKDC as a new key gene driving tumor heterogeneity.

## 4. Materials and Methods

### 4.1. Repair Gene Database Setup

The genes involved in different routes of DNA damage repair were identified through the KEGG pathway finder [37]. The following maps related to DNA repair and recombination proteins (ko03400) were used: base-excision repair (map03410), nucleotide-excision repair (map map03420), mismatch repair (map03430), DSB repair homologous recombination (map03440), and DSB repair non-homologous end joining (map03450). The selected DNA-repair genes used in this study are listed in Appendix A.

### 4.2. Mutation Database Setup

A total of 433 colon adenocarcinoma and 158 rectal adenocarcinoma patients were included in the analysis. Mutations identified with Mutect2 were downloaded from the TCGA repository. Mutations were filtered based on the judgement system implemented in Mutect2, as well as additional filters, including a minimum of 50× coverage with at least 5× coverage of the mutant reads. Mutation frequency was calculated based on the number of mutant reads and the total coverage of the locus. Overall mutation burden for each sample was the sum of mutations accepted by the applied filters. To validate the results, we performed cross analysis using three additional datasets obtained from the cBioPortal [38] repository. The analysis was performed using the TCGA provisional (*n* = 431), DFCI [39] (*n* = 619), and the Genentech [40] (*n* = 72) colorectal adenocarcinoma datasets.

### 4.3. Calculating Correlation between Patient Mutation Burden and Repair Gene Statuses

To identify repair genes linked to higher mutation burden, we calculated the correlation between mutation status of each gene and the overall mutation burden in all samples using the Mann–Whitney U test. The test was performed in the R environment using the *wilcox.test()* function. By this, we identified DNA-repair genes for which a mutation was linked to an increased overall mutation burden in these samples (Figure 1A). We performed the analysis using the mutation status of single genes, as well as the combination of multiple genes involved in a DNA repair mechanism. In this, a mutation could be present in any of the genes involved in the given repair mechanism. Samples were split based on the mutation status of a DNA-repair gene and mutation burden was compared between the two groups using a non-parametric test. To summarize the mutation accumulation timeline for each patient, mutations were represented using histograms, where the y-axis denotes the number of mutations in a given bin and the x-axis represents the mutation frequency based on percentile bins. In the case of selected repair gene mutations, the gene symbol of the repair gene is shown above the histogram.

### 4.4. Identifying DNA-Repair Associated Founder Mutations

Since mutations in DNA-repair genes occurred in high rates with variable mutation frequencies, we aimed to identify genes where a high mutation prevalence was present before the start of accumulating genetic alterations. In principle, the mutation proportion of the investigated gene is higher than the subsequent mutations because this particular mutation will be present in all descendent cells. We termed these as founder mutations in case at least half the subsequent mutations were within a 30% mutation frequency range following the given founder gene mutation.

### 4.5. Cell Culture Setup

The HCT116 cell line used in this study was purchased from the European Collection of Cell Cultures (ECACC, Salisbury, UK). Cells were cultured in McCoy’s 5A medium (Gibco, Life Technologies, Carlsbad, CA, USA) supplemented with 50 μg/mL penicillin-streptomycin (Gibco) and 10% fetal bovine serum (Gibco). Cells were maintained at 37 °C in a humidified incubator with 5% CO_2_ atmosphere.

### 4.6. Mutagenesis Experiments

Cells were subjected to 20 J/m^2^ ultraviolet-C (UV-C) light irradiation, then were grown for 48 h either in the absence or presence of 2 μM NU7441 PRKDC inhibitor (iPRKDC) (also known as KU-57788) (Selleck Chemicals, Munich, Germany) [41]; or 20 μM KU-55933 (Selleck Chemicals) [42], inhibitor of ATM (iATM); or both compounds. The PRKDC inhibitor NU7441 is routinely used in the literature and is a well-characterized specific small molecular compound to counteract PRKDC action [43,44,45]. After drug treatment, the medium was changed to fresh medium without drugs, allowing recovery for an additional 48 h. Altogether, five rounds of treatments (UV with or without the drugs) were performed in combination with recovery periods. Non-treated cells were handled in parallel without either UV irradiation or the addition of any drug. 

Multi-cell clones were isolated by limiting dilution and grown first on 96-well, then on 24-well plates to obtain colonies originating from approximately three individual cell clones. There were no phenotypic differences observed between the colonies grown from either treated or non-treated cells with regard to colony size and proliferation. Finally, cells were harvested by trypsinization and cell pellets were combined in order to have pooled samples grown from approximately ten individual cell clones. For sample collection, four samples were collected in the case of non-treated cells and three samples were collected from each of the following treatments: UV; UV + iPRKDC; UV + iATM; and UV + iPRKDC + iATM. Genomic DNA was subjected to whole-exome sequencing.

### 4.7. Analysis of Whole Exome Sequencing Data

Data analysis was performed in the Galaxy platform [46]. Raw sequencing data were checked using FASTQC quality control tool (http://www.bioinformatics.babraham.ac.uk/projects/fastqc, accessed on 19 October 2021), and sequences with average base quality of less than 20 and a minimum length shorter than 20 were trimmed using Trimmomatic [47].

The alignment of paired-end reads to the reference genome GRCh38 was done using Bowtie2 [48] with default settings. The aligned reads with a MAPQ score of less than 20 and PCR-duplicate reads were filtered using the Picard toolkit (https://broadinstitute.github.io/picard/, accessed on 19 October 2021). 

Mutations were identified with Mutect2 in the Genome Analysis Toolkit [49] with default settings applied. The number of mutations in normal and UV-treated samples were compared using the Mann–Whitney U test. The test was performed in the R environment using the *wilcox.test()* function.

### 4.8. Construction of Gene Expression Database

Microarray gene expression data of colorectal cancer patients were obtained from the NCBI Gene Expression Omnibus (GEO) database (http://www.ncbi.nlm.nih.gov/geo/, accessed on 14 January 2018). Array data files were processed in the R environment (http://www.r-project.org, accessed on 19 October 2021). The entire database contains 2110 samples from 15 datasets measured with the Affymetrix Human Genome U133A (GPL96) or the Human Genome U133 Plus 2.0 (GPL570) microarrays. Array quality control was performed for all samples using the “yaqcaffy” (https://www.bioconductor.org/packages/release/bioc/html/yaqcaffy.html, accessed on 19 October 2021) library. In this step, we checked the background, raw Q, percentage of present calls, presence of BioB-/C-/D- spikes, GAPDH 3′ to 5′ ratio, and the beta-actin 3′ to 5′ ratio. Gene chips were normalized with the MAS5 algorithm using the “affy” (http://bioconductor.org/packages/release/bioc/html/affy.html, accessed on 19 October 2021) library [50]. After the normalization, we retained only probes measured on both GPL96 and GPL570 platforms (*n* = 22,277). Then, a second scaling normalization was applied to set the mean expression on each chip to 1000. For genes measured by several probe sets, we used JetSet to select the most reliable probe set [51].

### 4.9. Analysis of Clinical Samples

Expression in normal and tumor tissues was compared using the Mann–Whitney test. We examined the correlation between the expression of the selected repair genes and relapse-free survival (RFS) using Cox proportional hazard regression and by plotting Kaplan–Meier survival plots. Cox regression analysis was performed using the “survival” R package v2.38 downloaded from CRAN (https://cran.r-project.org/web/packages/survival/index.html, accessed on 19 October 2021). Kaplan–Meier plots were generated for each gene separately and for the mean expression of genes involved in functional groups of DNA-repair genes by applying the “survplot” R package v0.0.7 (http://www.cbs.dtu.dk/~eklund/survplot/, accessed on 19 October 2021).

## Figures and Tables

**Figure 1 ijms-23-00633-f001:**
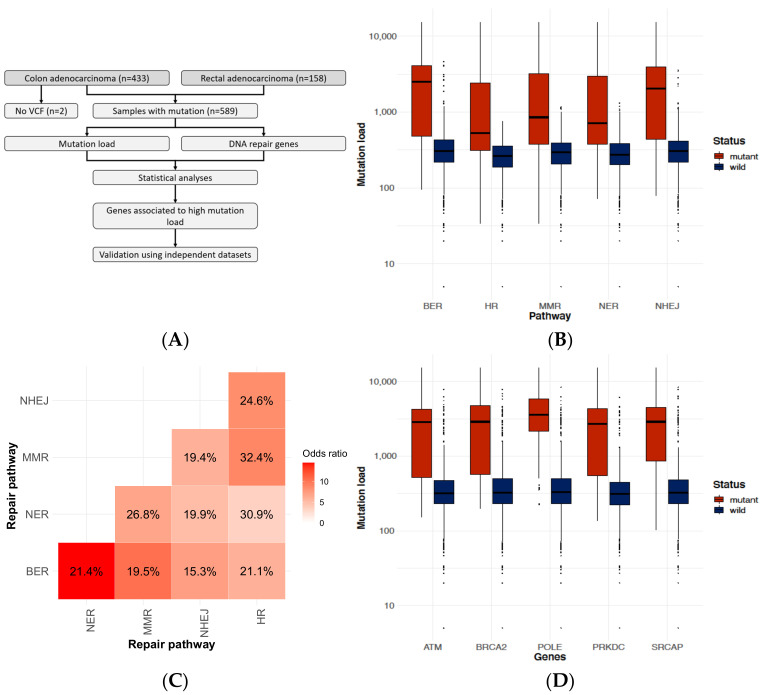
Characteristics of DNA repair pathway mutations in colorectal cancer. Summary of the analysis pipeline of selecting DNA-repair genes linked to increased mutation burden in colorectal cancer samples of the TCGA repository (**A**). A higher mutation load (median and interquartile ranges of the total mutation count) was observed when mutations were present in any of the DNA repair pathways (**B**). Simultaneous mutations in DNA repair pathways show enrichment for each signature as demonstrated by the high odds ratios of co-mutation when comparing DNA repair mechanisms summarized in a pairwise heatmap format. Percentage of patients with co-mutation in pathways is displayed within the tiles of the heatmap (**C**). Mutation load (boxplots showing median values with lower and upper quartile values and whiskers marking the range) related to mutation in most commonly mutated genes associated with DNA repair (**D**). BER = base excision repair, HR = homologous recombinational repair, MMR = mismatch repair, NER = nucleotide excision repair, and NHEJ = non-homologous end joining repair.

**Figure 2 ijms-23-00633-f002:**
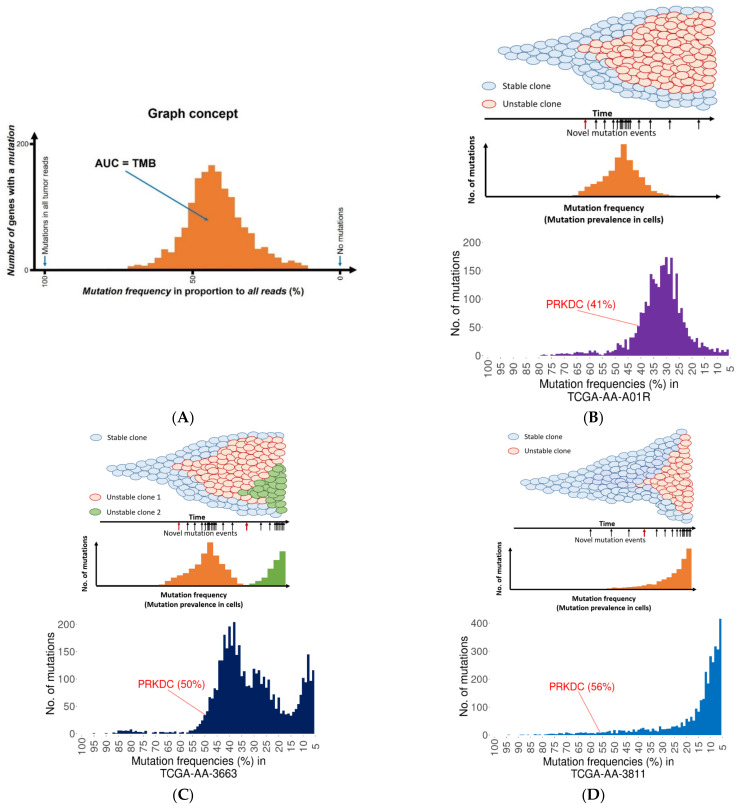
Founder mutations lead to an increased mutation load. Theoretical histogram showing the number of genes with a mutation as a function of the mutation frequency in a sample (**A**). The area under the curve (AUC) equals the total mutation burden (TMB) in a sample. Theoretical and actual representation using a representative example (with TCGA sample identifier) of founder mutation and different mutation load patterns in colorectal cancer patients with a single (**B**), double (**C**), and exponential (**D**) clonal expansion. The red arrows show the incidence of the founder mutation. Mutations in the *PRKDC* gene are indicated in the TCGA samples.

**Figure 3 ijms-23-00633-f003:**
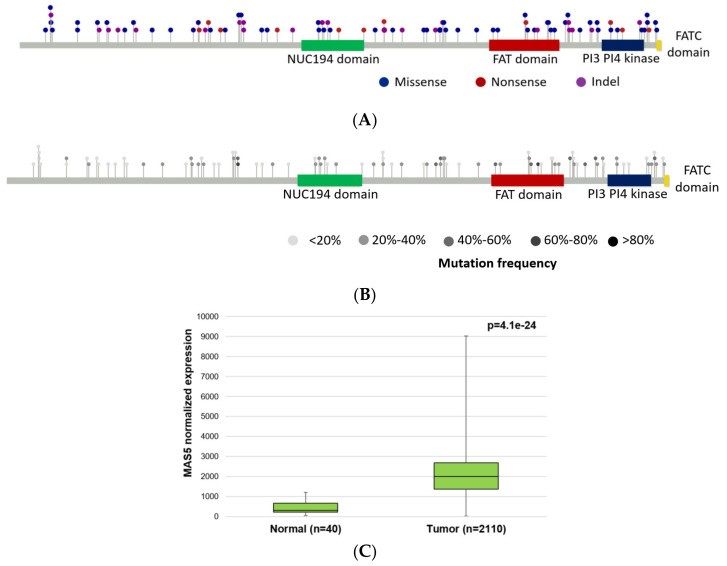
Localization of *PRKDC* mutations show uniform distribution in the entire gene using lollipop plot figures representing mutations in an information-dense manner indicating the positions in amino acid coordinates within the domain organization scheme of the gene (**A**,**B**). Some mutations are localized in annotated domains of *PRKDC* (NUC194, FAT, and PI3–PI4 kinase domains), while other mutations are localized in structurally/functionally unknown protein segments (grey line). Distribution of different types of mutations color coded in *PRKDC* (**A**) and mutation frequency for each sample using different shades of grey (**B**). Expression of *PRKDC* is significantly higher in colorectal cancer samples (microarray gene expression data from the NCBI Gene Expression Omnibus (GEO) database, see Section 4.8) (**C**).

**Figure 4 ijms-23-00633-f004:**
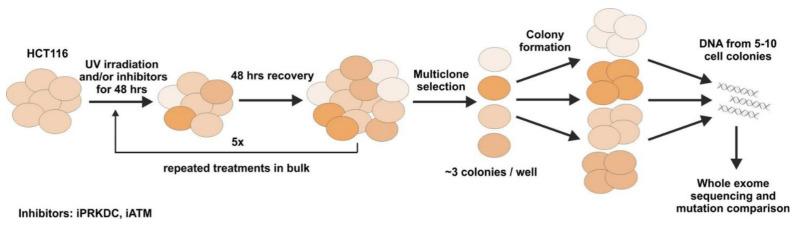
Experimental overview of the in vitro experiments. Cell lines were treated with UV to induce DNA damage and inhibitors against *PRKDC* and *ATM*. Five rounds of treatment with recovery periods were applied. Multiclone cell colonies (approx. three colonies/well) were cultured in 96-well plates. DNA was extracted from approximately 10 colonies, creating a polyclonal mixture, where drug-induced mutations were amplified over detection thresholds. Samples were subjected to whole-exome sequencing for mutation analysis.

**Figure 5 ijms-23-00633-f005:**
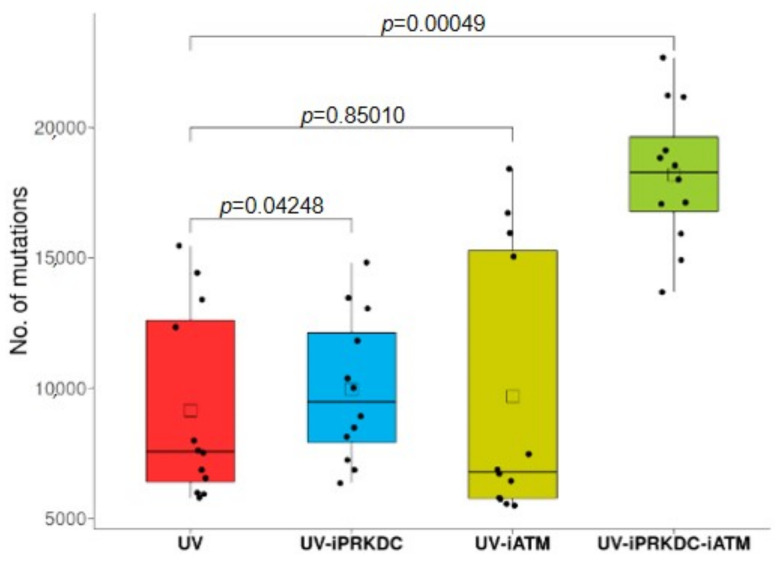
The number of mutations show a significant increase in samples treated with UV and either *PRKDC* or both inhibitors. Boxplot showing the number of mutations including SNVs and indels in the samples treated with UV, iPRKDC, iATM, or both. The mean numbers of mutations in each category are represented by squares.

**Table 1 ijms-23-00633-t001:** Top genes associated with high mutation burden in colorectal cancer.

Cohort	Gene	*p*-Value	Mutation Burden (Mutant Samples)	Mutation Burden (Wild Samples)	Fold Increase	Samples with Mutation (%)	Total Samples
TCGA COAD (mutect2)	* **PRKDC** *	**9.5 × 10^−28^**	**2853**	**329**	**8.7**	**22.0**	**431**
	*ATM*	3.7 × 10^−24^	2926	333	8.8	20.0	431
	*POLE*	1.3 × 10^−22^	4002	336	11.9	13.9	431
	*BRCA2*	1.2 × 10^−20^	2951	335	8.8	15.1	431
	*POLD1*	5.1 × 10^−18^	3538	348	10.2	10.2	431
DFCI	*POLE*	2.2 × 10^−14^	919	123	7.5	7.3	619
	*BRCA2*	1.2 × 10^−13^	948	123	7.7	6.3	619
	*ATM*	1.4 × 10^−12^	819	123	6.7	7.4	619
	* **PRKDC** *	1.3 × 10^−9^	**765**	**124**	**6.2**	**6.8**	**619**
	*MLH3*	1.1 × 10^−8^	1122	126	8.9	3.2	619
GenenTech	*ATM*	4.4 × 10^−4^	909	82	11.1	18.1	72
	*RAD50*	1.2 × 10^−3^	2393	86	27.8	5.6	72
	*BRCA2*	1.5 × 10^−3^	1218	84	14.5	6.9	72
	* **PRKDC** *	2.2 × 10^−3^	**1671**	**86**	**19.4**	**5.6**	**72**
	*LIG1*	3.2 × 10^−3^	1195	84	14.2	6.9	72

**Table 2 ijms-23-00633-t002:** Results of the survival analysis for the expression of the top genes associated with high mutation burden in colorectal cancer patients. RFS = relapse-free survival, CI = 95% confidence interval.

Gene	HR	CI	*p*-Value	Expression Fold Change (Tumor vs. Normal)
*PRKDC*	0.72	0.58–0.9	4.40 × 10^−3^	6.44
*BRCA2*	0.73	0.58–0.91	5.60 × 10^−3^	4.30
*RAD50*	0.81	0.64–1.02	6.90 × 10^−2^	2.74
*ATM*	1.18	0.93–1.49	1.68 × 10^−1^	1.48
*LIG1*	1.17	0.94–1.46	1.70 × 10^−1^	1.40
*MLH3*	1.48	1.13–1.95	4.70 × 10^−3^	1.38
*POLD1*	0.67	0.53–0.85	8.0 × 10^−4^	1.09
*POLE*	0.61	0.47–0.77	5.30 × 10^−5^	0.95

## Data Availability

Not applicable.

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
