# Peer review of "Primary Founder Mutations in the PRKDC Gene Increase Tumor Mutation Load in Colorectal Cancer"

_ijms, 2022, doi:10.3390/ijms23020633_

Round 1
Reviewer 1 Report
The manuscript is well written and structured. The authors performed a unique analysis of the correlation between mutation load and the DNA-repair transcripts' expression level in colorectal cancer of different grades. The patient survival data in relation to the mean expression of all DNA-repair genes are also provided. Of the 180 genes involved in DNA repair, the top five were described as the mutation founders. Detailed experiments were performed with an accent on the inhibition of PRKDC and ATM1 transcripts in relation to the UV-induced mutation rate. Overall, this is an interesting and well-designed study, which I recommend for publication in the Int. J. Mol. Sci with minor revision. Bellow I have provided a couple of suggestions on experimental setting:
- Figure 1D. Would you please specify what values are presented in the graph as the mutation load (median of the total mutation count or average+/-SD of the total mutation count)?
- P7. Would you please describe the colony selection process? Please provide proliferation/or viability values for colonies or cell culture overall in the presence of the PRKDC, ATM1, and PRKDC plus ATM1 inhibitors versus control treatment in UV experiment on HCT116 cell line. Also, the information about the relative size of the selected colonies at different treatment conditions versus control will be helpful.
Reviewer 2 Report
Palinkas et al have analyzed 591 colorectal tumors and looked at overall mutation load as well as the mutation status of DNA repair genes in these tumors. They identified founder mutations in a few key DNA repair genes, including PRKDC. They also observed an overall increase in the PRKDC expression in tumor compared to normal samples. Overall, their results indicate PRKDC as a new key gene in driving tumor heterogeneity.
Some of the data representation needs to be improved significantly. The rationale behind some of the experiments needs to be addressed in a more comprehensive manner. The way the experiments have been conducted need to be explained in better detail. The manuscript needs some major updates and changes before it can be accepted for publication. Some of my major comments are outlined below.
- The biggest concern with this manuscript is the designation of founder mutations. The authors report that mutations that have the highest prevalence across all patients have been designated as founder mutations. However, this is an inaccurate way of designating founder mutations. Just because these mutations are more prevalent does not mean that they are founder mutations. Some of the highest prevalent mutations may have been induced as a result of critical mutations in some DNA editing enzymes making them hyperactive. E.g a large number of mutations earlier thought of as driver mutations due to their high prevalence and frequent occurrence in patients are actually passenger mutations that are induced by APOBEC enzymes.
- Moreover, since the authors have analysed patient data, did the patients receive any chemotherapy treatment earlier? It is an important point that needs to be addressed which can change the outcome of their interpretation. If there are some patients who have previously received chemotherapy, their mutation burden is going to be vastly different than others who have not and will have a high prevalence of mutations.
- One of the conclusions is that PRKDC expression is increased in tumors samples compared to normal samples. But high expression leads to longer relapse free survival. These two points are counter-intuitive and need to be addressed properly in the text with better explanation.
- The introduction seems to have an abrupt ending. There should be statements toward the end of the introduction stating the gap and what the authors have tried to do to address this gap and couple of their main conclusions.
- It would be helpful if the authors indicate what datasets they are using for the analysis right at the start of the results section instead of in the next section.
- 1B: How did the authors define mutations in DNA repair mechanisms? Which proteins in those pathways were mutated? This should be mentioned for Fig.1B.
- 1C: the authors analyzed whether mutation in one DNA repair pathway resulted in increased mutation prevalence in other repair pathway and found that when one of the pathways was mutated, another hit in an alternative pathway was more common. How did they identify this? How do they analyse whether the mutations are dependent or have occurred in parallel?
- 1C: This figure should be explained a bit better. What do those numbers mean? I assume they are the individual tumors that the authors analysed? Why does the sum not add up to 591 but only to 431. Why have the authors excluded 160 tumors from this analysis. This result is correlative and should not be used to conclude that mutations in pathway are dependent on those in the other pathways.
- Page 3: Line 94-96. The rationale does not quite fit right. As mentioned above, just because the mutation is more prevalent does not necessarily mean it had developed earlier. There may have been some events later in tumor evolution that mutated those subset of genes with a very high frequency.
- Fig 3: This figure is very confusing and again needs to be explained very carefully.
- In the mutagenesis experiments, what is the rationale for using HCT116 cells? These cells are great to work with but may not be a suitable model for the purpose of the experiments. These cells are mismatch repair deficient and also have mutations that lead to a reduced expression of MRE11. These deficiencies in DNA repair factors can greatly affect how the mutagenesis experiment is carried out. It is advisable to use other cell lines to perform this assay.
- The discussion section is well written but needs additional explanation on the results and what do they mean in the context of the current knowledge.
Round 2
Reviewer 2 Report
I am happy with all the changes made by the authors. I recommended accepting the manuscript for publication.